# Defining the Relative Role of Insulin Clearance in Early Dysglycemia in Relation to Insulin Sensitivity and Insulin Secretion: The Microbiome and Insulin Longitudinal Evaluation Study (MILES)

**DOI:** 10.3390/metabo11070420

**Published:** 2021-06-26

**Authors:** Alexis C. Wood, Elizabeth T. Jensen, Alain G. Bertoni, Gautam Ramesh, Stephen S. Rich, Jerome I. Rotter, Yii-Der I. Chen, Mark O. Goodarzi

**Affiliations:** 1USDA/ARS Children’s Nutrition Research Center, Baylor College of Medicine, Houston, TX 77030, USA; Alexis.Wood@bcm.edu; 2Department of Epidemiology and Prevention, Wake Forest School of Medicine, Winston-Salem, NC 27101, USA; ejensen@wakehealth.edu (E.T.J.); abertoni@wakehealth.edu (A.G.B.); 3School of Medicine, University of California, La Jolla, San Diego, CA 92093, USA; gramesh@health.ucsd.edu; 4Center for Public Health Genomics, University of Virginia, Charlottesville, VA 22908, USA; ssr4n@virginia.edu; 5Institute for Translational Genomics and Population Sciences, The Lundquist Institute for Biomedical Innovation and Department of Pediatrics, Harbor-UCLA Medical Center, Torrance, CA 90502, USA; jrotter@lundquist.org (J.I.R.); ichen@lundquist.org (Y.-D.I.C.); 6Division of Endocrinology, Diabetes, and Metabolism, Department of Medicine, Cedars-Sinai Medical Center, Los Angeles, CA 90048, USA

**Keywords:** insulin sensitivity, insulin secretion, insulin clearance, disposition index, prediabetes, diabetes

## Abstract

Insulin resistance and insufficient insulin secretion are well-recognized contributors to type 2 diabetes. A potential role of reduced insulin clearance has been suggested, but few studies have investigated the contribution of insulin clearance while simultaneously examining decreased insulin sensitivity and secretion. The goal of this study was to conduct such an investigation in a cohort of 353 non-Hispanic White and African American individuals recruited in the Microbiome and Insulin Longitudinal Evaluation Study (MILES). Participants underwent oral glucose tolerance tests from which insulin sensitivity, insulin secretion, insulin clearance, and disposition index were calculated. Regression models examined the individual and joint contributions of these traits to early dysglycemia (prediabetes or newly diagnosed diabetes). In separate models, reduced insulin sensitivity, reduced disposition index, and reduced insulin clearance were associated with dysglycemia. In a joint model, only insulin resistance and reduced insulin secretion were associated with dysglycemia. Models with insulin sensitivity, disposition index, or three insulin traits had the highest discriminative value for dysglycemia (area under the receiver operating characteristics curve of 0.82 to 0.89). These results suggest that in the race groups studied, insulin resistance and compromised insulin secretion are the main independent underlying defects leading to early dysglycemia.

## 1. Introduction

Type 2 diabetes is a leading cause of blindness, kidney failure, and amputation worldwide, and is associated with a high risk of coronary heart disease and stroke. Of note, even people with impaired fasting glucose (IFG) or impaired glucose tolerance (IGT) (conditions considered prediabetes) are at an increased risk of cardiovascular disease. Therefore, physiologic studies of dysglycemia (prediabetes and diabetes) are critically needed to improve public health, as roughly a quarter of the US population has either prediabetes or diabetes [1].

Insulin homeostasis encompasses the production of insulin by pancreatic beta cells (insulin secretion), its metabolic effects on skeletal muscle, liver, and adipose tissue (insulin sensitivity), and its removal from the circulation by the liver, kidney, and peripheral tissues (insulin clearance). The pathophysiology of type 2 diabetes involves defects in multiple components of insulin homeostasis. A leading theory is that impaired insulin sensitivity (i.e., insulin resistance) is the earliest defect, related in large part to unhealthy lifestyle, but also genetic factors. In order to increase circulating insulin levels, the insulin homeostatic system responds to insulin resistance by increasing insulin secretion and reducing insulin clearance. If this hyperinsulinemic compensation is adequate, circulating glucose levels will remain within normal limits [2,3]. Failure of insulin secretion to overcome insulin resistance is regarded by many as the key event leading to type 2 diabetes [4]. This concept is supported by the observation that the majority of genetic loci implicated in type 2 diabetes appear to act via compromising beta cell development or function [5]. In contrast to the substantial body of work focused on insulin secretion, relatively fewer studies have examined the role of insulin clearance in the development of diabetes.

The hepatic clearance of insulin plays a critical role in the bioavailability of insulin; after pancreatic secretion of insulin into the portal circulation, the liver is the first major organ to encounter insulin. Insulin that is not extracted from the circulation by the liver then reaches the rest of the body, where it can exert its effects to lower blood glucose. Estimates are that 80% of endogenously secreted insulin is cleared by the liver (~50% at first pass); ~15% is cleared by the kidney; and ~5% by muscle and adipose tissue [6].

Insulin clearance and insulin sensitivity have a well-established positive correlation [7,8,9,10,11,12,13]. Several lines of evidence suggest that reduced insulin clearance is a compensatory response to insulin resistance. When fed a high-fat diet, dogs developed increased visceral adiposity and insulin resistance, which evoked increased insulin secretion and reduced insulin clearance [2,14]. However, reduced insulin clearance may exacerbate insulin resistance via hyperinsulinemia-mediated desensitization [15,16]. Diminished insulin clearance at baseline has been found to correlate with future development of diabetes [9,13,17]. A hypothesis has emerged that reduced insulin clearance is a major contributor to insulin resistance and therefore represents an independent risk factor for diabetes [18,19].

Given the physiologic interrelationships between insulin sensitivity, insulin secretion, and insulin clearance, any analysis attempting to assess whether one of these is a predictor of diabetes must account for joint or confounding effects of the other two. To date, few studies examining the role of insulin clearance in diabetes have simultaneously considered insulin sensitivity or insulin secretion. In the Microbiome and Insulin Longitudinal Evaluation Study (MILES), all three traits were estimated by multi-point oral glucose tolerance testing (OGTT) [20]. Therefore, in the current study, we assessed the correlation of insulin clearance with early dysglycemia (prediabetes and newly recognized diabetes) in the context of insulin sensitivity and insulin secretion. We found that when considered together, insulin sensitivity and insulin secretion, but not insulin clearance, were independent predictors of dysglycemia.

## 2. Results

We assessed the correlation of insulin clearance with the other insulin homeostasis traits (Figure 1). Insulin clearance exhibited a strong positive relationship with insulin sensitivity and a negative correlation with insulin secretion. Insulin clearance was not associated with disposition index (the product of insulin sensitivity and insulin secretion).

Though the primary focus of the current study is to compare individuals with and without dysglycemia, Table 1 and Figure 2 display phenotypic characteristics between those with normal glucose tolerance, prediabetes, and diabetes to illustrate trends that may characterize disease progression. Notably, there were no differences in race between glycemic categories. As expected, glucose, both fasting and at 2 h of the OGTT, progressively increased between worsening categories of glycemia. Stepwise increases were also observed for the waist–hip ratio and C-peptide, while progressive decreases were seen in insulin sensitivity and disposition index. Insulin secretion decreased mainly between prediabetes and diabetes, whereas insulin clearance was reduced in prediabetes compared to normal glucose tolerance (Figure 2).

Table 2 presents the insulin homeostasis traits according to dysglycemia status, wherein those with prediabetes and diabetes are combined. Insulin sensitivity, disposition index, and insulin clearance were all lower in dysglycemia. Insulin secretion did not differ between normal glucose tolerance and dysglycemia, largely reflecting the proportion of the latter group with prediabetes (83% of the dysglycemic group), in whom insulin secretion was not different than those with normal glucose tolerance (Figure 2). When stratified by race, the same differences in insulin homeostasis traits were observed, except that insulin clearance was not significantly different between African Americans with and without dysglycemia (Appendix A).

Given that most of the dysglycemic group had prediabetes, we also compared insulin homeostasis traits in those with normal glucose tolerance to those with prediabetes (IFG and/or IGT), IFG, or IGT. As for the dysglycemic group, the groups with prediabetes or IGT had lower insulin sensitivity, disposition index, and insulin clearance, but similar insulin secretion compared to the group with normal glucose tolerance (Appendix A). Those with IFG had lower values of all four insulin homeostasis traits (Appendix A).

To assess the ability of the insulin homeostasis traits to predict dysglycemia, we started with a base model consisting of age, sex, body mass index (BMI), and race. This base model had an area under the receiver operating characteristics curve (AUROC) of 0.68, with age, sex, and BMI being independent correlates of dysglycemia in the model (Table 3). We then constructed models where each of the four insulin homeostasis traits were individually added to the base model. The AUROC was significantly increased compared to the base model in models that included insulin sensitivity or disposition index (Table 4, Figure 3). In contrast, addition of insulin clearance or insulin secretion to the base model did not result in any change in the discriminative value of the model.

In the base model and models 1–4 and model 6, variance inflation factors (VIF) ranged from 1.01 to 1.66. In model 5, VIF values were 1.14, 1.05, 1.66, 1.32, 3.00, 2.59, and 3.46 for age, sex, BMI, race, insulin sensitivity, insulin secretion, and insulin clearance, respectively.

Next, to assess the independent contribution of the features of insulin homeostasis to dysglycemia, we constructed a model consisting of the base model plus insulin sensitivity, insulin secretion, and insulin clearance (model 5). In this joint model, reduced insulin sensitivity, and reduced insulin secretion were independent predictors of dysglycemia, while insulin clearance was not significant. The AUC of this model was significantly different than the base model plus insulin sensitivity. While the AUROC value of the base model plus three insulin homeostasis traits was marginally (*p* = 0.05) different than that of the base model plus disposition index, the difference in Akaike’s information criterion indicated that the three-trait model was superior in discriminating dysglycemia from normal glycemia. Modest collinearity was observed in the model with three insulin traits (variance inflation factors (VIF) of 3.0, 2.6, and 3.5 for insulin sensitivity, insulin secretion, and insulin clearance, respectively). In response to this, we examined a model containing both insulin clearance and DI (model 6), under the assumption that DI is an integrated measure of insulin sensitivity and insulin secretion. In this model, which exhibited acceptable VIF values, insulin clearance was significantly associated with dysglycemia; however, the AUROC was not improved compared to the model with DI (Table 4). Essentially the same results were observed in analyses focused on prediabetes, IGT, or IFG (Appendix A).

We repeated the above analyses separately in African Americans and non-Hispanic Whites (Appendix A), wherein the base model consisted of age, sex, and BMI. Results in non-Hispanic Whites (Appendix A) closely mirrored those of the entire group (Table 3). Results in African Americans were also similar with the exception that insulin clearance was not significantly associated with dysglycemia in model 3; however, similar to the entire group, insulin clearance and disposition index were significantly associated with dysglycemia in model 6 (Appendix A). AUROC comparisons within each of the two race groups (Appendix A) yielded similar results as the primary analysis (Table 4).

## 3. Discussion

In this study featuring all three aspects of insulin homeostasis, we found that while decreased insulin clearance was associated with an increased prevalence of early dysglycemia, the correlation between clearance and dysglycemia was no longer significant when jointly analyzed with insulin sensitivity and insulin secretion. On the other hand, insulin sensitivity and insulin secretion remained as significant independent predictors of dysglycemia, consistent with the prevailing notion that insulin resistance and deficient insulin secretion are key to the pathogenesis of diabetes. Further consistent with this notion, reduced insulin sensitivity and reduced disposition index were highly predictive of dysglycemia.

It has been well-documented that insulin clearance is lower in states of obesity, prediabetes, and diabetes [21]; however, most studies reporting this did not attempt to adjust for effects of insulin sensitivity or insulin secretion. The current results highlight the importance of considering effects of insulin sensitivity and insulin secretion when studying insulin clearance. Insulin clearance has a strong positive relationship with insulin sensitivity [7,8,9,10,11,12,13] and an inverse relationship with insulin secretion [13]. Given that reduced insulin sensitivity (i.e., insulin resistance) is a risk factor for diabetes, a univariate analysis finding apparent association between reduced insulin clearance and diabetes may be driven by the confounding effect of insulin resistance, rather than a primary effect of insulin clearance. In the Insulin Resistance Atherosclerosis Study (IRAS) Family Study (1116 Hispanic and African American participants), we found that reduced insulin clearance (measured by the frequently sampled intravenous glucose tolerance test (FSIGT)) at baseline was associated with higher risk of incident diabetes at 5 years of follow-up, including after adjustment for insulin secretion [9]. However, when the analysis was further adjusted for insulin sensitivity, the association of reduced clearance with incident diabetes was no longer significant, consistent with the current results in MILES. The Prospective Metabolism and Islet Cell Evaluation (PROMISE) cohort followed 492 individuals (71% European, 29% other) over 9 years, finding that decreased baseline or declining insulin clearance (measured by OGTT using the same index used in MILES) was associated with incident dysglycemia (also defined as in MILES) [17]. When insulin sensitivity was included in the models, baseline insulin clearance was no longer associated with incident dysglycemia while the association with longitudinal change in insulin clearance (by AUC-Cpep/AUC-Ins) was attenuated but remained significant [22]. A modeling study using hyperglycemic and euglycemic–hyperinsulinemic clamps in Japanese subjects with normal glucose tolerance, impaired glucose tolerance, and diabetes found that insulin clearance progressively diminished in these three groups, suggesting that insulin clearance predicts the progression of glucose intolerance. However, this study did not dissect whether this was independent of insulin sensitivity or secretion [11]. In contrast to our results, a recent study of Southwestern Native Americans (a population with an extremely high risk of diabetes) did simultaneously analyze the components of insulin homeostasis (measured using euglycemic clamps and intravenous glucose tolerance tests) and found that all three were independent predictors of incident diabetes, which occurred in 32% of the cohort [13]. Whether reduced clearance independently predicts diabetes in other racial and ethnic groups with lower risk of diabetes remains to be seen. Our results in the IRAS Family Study and MILES suggest this may not be the case in African Americans, Hispanics, or non-Hispanic Whites. This is noteworthy given that numerous studies have found that insulin clearance is decreased in individuals of African descent, compared to other racial groups, and is a driver of hyperinsulinemia [23,24,25].

The direction of causality between insulin resistance and reduced insulin clearance has been debated. Studies in mice deficient in Ceacam1, one of the best characterized proteins involved in insulin clearance, suggest that diminished clearance promotes insulin resistance [16,26]. However, mouse models where the function of a key molecule has been manipulated may not reflect human physiology. Furthermore, interventions that improve insulin sensitivity (rosiglitazone [27], weight loss [8], exercise [28]) were found to increase insulin clearance, suggesting that insulin resistance suppresses insulin clearance. While the underlying molecular mechanisms remain to be identified, reduced insulin clearance appears to be an early adaptive response to insulin resistance [29,30]. Reduced insulin clearance is likely to be metabolically beneficial in insulin-resistant states, as it elevates plasma insulin levels without increasing the secretory burden on pancreatic β-cells. Consistent with this idea, administration of salsalate, an inhibitor of insulin clearance, improved glucose homeostasis in insulin resistant individuals, whereas it had no effect on insulin action [31]. In high-fat fed dogs who developed insulin resistance, reduced insulin clearance was more important in maintaining long-term hyperinsulinemia than increased insulin secretion [3]. In the context of these studies, our results suggest that as a compensatory factor, reduced insulin clearance may be a biomarker for diabetes, but not a primary causal factor. Though apparently not a main independent causal factor such as insulin resistance or deficient insulin secretion, the observed interindividual variability [32] of insulin clearance in the population may be a contributing determinant of an individual’s capacity for hyperinsulinemic compensation and thus risk for diabetes. A possible role is suggested by our finding that the model with all three insulin homeostasis traits had the highest AUROC, as well as insulin clearance being significant in the model with both insulin clearance and DI. The role of insulin clearance may be more significant in high-risk groups such as Native Americans [13].

Another possible explanation of our results is that reduction in insulin clearance is a robust compensatory mechanism that does not initially fail, unlike beta-cell compensation, whose failure is considered a key event in the development of diabetes. However, after diabetes has developed, increasing clearance would reduce the amount of insulin available to promote cellular glucose uptake, and thereby may be an independent predictor of worsening glycemia over time, as demonstrated in a prospective study of 732 patients with recently diagnosed type 2 diabetes [33]. This suggests that sequential failure of insulin homeostasis mechanisms contribute to the onset of diabetes (failure of insulin secretion to remain high) and potentially to the progression of diabetes (failure of insulin clearance to remain low). Emerging evidence suggests that there is substantial heterogeneity within individuals diagnosed with type 2 diabetes. Active efforts are underway to use physiologic and genetic parameters to define subtypes of diabetes [34,35]. It is conceivable that there is a subset of people who develop diabetes due to insufficient compensatory reduction of insulin clearance in response to insulin resistance, while a different subset develops insulin resistance due to hyperinsulinemia caused by low insulin clearance. In certain racial groups (e.g., South Asians), the primary defect may be deficient insulin secretion [36]. Thus, three subtypes of diabetes may exist, wherein the initial defect resides in each component of insulin homeostasis (resistance, secretion, clearance).

While both insulin resistance and decreased insulin secretion were independent correlates of dysglycemia, we found a significantly higher predictive value for insulin resistance. A recent study that jointly modeled the three aspects of insulin homeostasis also found that insulin resistance had a greater magnitude of effect on incident diabetes than decreased insulin secretion and decreased insulin clearance [13]. Given that our dysglycemic group is early in the trajectory of abnormal glucose regulation (prediabetes or newly recognized diabetes), this difference suggests that insulin resistance may be the earliest defect on the path to diabetes. Consistent with this notion, a comparison of those with normal glucose tolerance, impaired glucose tolerance, and type 2 diabetes found that insulin sensitivity declined between normal and impaired glucose tolerance whereas insulin secretion was decreased from impaired glucose tolerance to diabetes [11]. In one of the largest studies (over 6500 participants) documenting the trajectories of insulin sensitivity and insulin secretion prior to the development of diabetes, insulin sensitivity exhibited a steep decline 5 years prior to the diagnosis of diabetes [37]. Insulin secretion initially rose in compensation 4 years prior to the diagnosis, but then started to decline 3 years prior to the onset of diabetes; insulin secretion remained higher than that of those who did not develop diabetes until immediately before diagnosis, at which point insulin secretion was markedly low. This large study provides additional evidence that insulin resistance may be the earliest defect and thus highly predictive of dysglycemia; unfortunately, the study did not assess insulin clearance.

Another possible explanation for the lower predictive value of insulin secretion compared to insulin resistance is the changing nature of insulin secretion in the progression of early dysglycemia. As noted above, in the years before diabetes onset, insulin secretion increases and then decreases [37]. We observed a similar trend in our data (Figure 2). Thus, the absolute level of insulin secretion may not consistently predict dysglycemia, especially in those with prediabetes, which constituted the majority of our dysglycemic group. The dynamic variation of insulin secretion is well-handled by the disposition index, which estimates the adequacy of the insulin secretion response to insulin resistance [38] and is highly predictive of diabetes, both in our data and studies of others [39,40]. Unlike insulin secretion, disposition index progressively decreases going from normal glucose tolerance to impaired glucose tolerance to diabetes [10,11], which we also observed in our study.

The current results suggest that an OGTT-based measurement of insulin sensitivity, disposition index, or all three aspects of insulin homeostasis, in combination with age, sex, BMI, and race, may be very effective in predicting future dysglycemia, with AUROC of 0.82–0.89. These high values suggest that the insulin homeostasis traits studied represent the most important physiological defects underlying early dysglycemia. These models may be useful in population screening efforts to identify individuals at highest risk, allowing targeted application of diabetes prevention efforts. Currently used screening in clinical settings relies mainly on fasting glucose and hemoglobin A1c measurements, which detect prediabetes and diabetes after they have occurred. In contrast, models based on insulin homeostasis traits developed herein could be used to predict early glycemia. Future large prospective studies will be needed to validate the value of these markers in the prediction of incident prediabetes and diabetes and to assess whether they have benefit compared to glycemia-based screening. Because insulin clearance requires C-peptide measurement, such studies could save resources by relying on the disposition index, which requires only glucose and insulin measurements during OGTT and had an AUROC (0.87, model 4) marginally lower than the model including three insulin traits (0.89, model 5).

A limitation of our study is that we used OGTT-derived measures of the components of insulin homeostasis, rather than direct measures such as those derived from clamp studies or the FSIGT. This is a particular concern regarding our measure of insulin clearance. Our measures of insulin sensitivity and insulin secretion have repeatedly demonstrated strong correlation with direct measures of these traits (r of 0.7–0.8) [41,42,43]. In addition, of 11 OGTT-based indices for insulin secretion, the index we used was most highly associated with diabetogenic single nucleotide polymorphisms implicated as acting via compromised insulin secretion [42]. On the other hand, insulin clearance is a challenging trait to measure even by detailed techniques, as FSIGT measures dynamic clearance while euglycemic clamp measure steady state clearance, yielding different results. Few studies have assessed the correlation of C-peptide to insulin ratios with directly measured insulin clearance. One small study (41 individuals who underwent both OGTT and euglycemic clamp) found the correlation to be high at 0.74 [44]. As euglycemic clamp yields whole body insulin clearance (infusion rate divided by steady state insulin), we are not aware of any studies correlating C-peptide to insulin ratio with specific measures of hepatic clearance. Given that insulin but not C-peptide is cleared by the liver on first pass, this ratio is widely used as a surrogate measure of hepatic insulin clearance. However, the ratio of C-peptide to insulin depends not only on their secretion, but also their kinetics of disappearance, which differ substantially [45]. Given the differing half-lives of C-peptide (30 min) and insulin (4 min), their ratio may not be an accurate estimation of hepatic insulin clearance [46]. Furthermore, hepatic insulin clearance and extrahepatic clearance may be independently regulated [32], and therefore may have differential impact on the risk of dysglycemia. Hepatic but not extrahepatic clearance was found to be associated with muscle and adipose tissue insulin resistance [47]. We eagerly await the future development of more accurate and specific OGTT-based markers of hepatic and extrahepatic insulin clearance. Once available, we would use those measures to re-evaluate the current analyses. All of our insulin homeostasis trait variables utilized the same insulin measurements. In an ideal setting, each trait would have been measured by independent phenotyping procedures, avoiding issues of collinearity. We also note that our dysglycemia outcome predominantly reflects prediabetes and early undiagnosed diabetes; therefore, the current results do not inform on the role of insulin homeostasis traits in diabetes of longer duration. We cannot rule out the possibility that a larger sample size would have been able to detect an independent effect of insulin clearance on dysglycemia. Finally, while providing insights regarding the physiologic deficits underlying early dysglycemia, our analyses do not prove causal relationships.

The current analysis is cross-sectional, based on the baseline data of MILES. As MILES is a prospective study, in the future we will be able to assess the predictive value of the components of insulin homeostasis on incident diabetes. It is anticipated that future prospective analyses in MILES and other cohorts may provide clarity regarding the role of reduced insulin clearance in the development of diabetes, including whether decreased clearance plays a general role or is diabetogenic only in specific groups such as Native Americans [13] or children with obesity [10].

## 4. Materials and Methods

### 4.1. Study Participants

Recruitment methods and eligibility criteria of the MILES cohort were previously described in detail [20]. The cohort consists of 353 individuals (129 African American and 224 non-Hispanic White) aged 40–80 living in the Piedmont Triad area of North Carolina. Three study visits are planned in MILES; the current analyses were performed using data collected at the first visit, which all subjects had completed. The presence of diabetes at baseline (by history or point of care fasting glucose ≥ 7.0 mmol/L) was an exclusion criterion; however, oral glucose tolerance testing (OGTT) performed during the study found that 25 individuals had 2-h glucose levels of 11.1 mmol/L or greater and three individuals had fasting glucose levels slightly greater than 7.0 mmol/L. These 28 were classified as having diabetes in the current analysis. None of these individuals were taking antidiabetic medication. An additional 136 participants were classified as having prediabetes based on IFG (5.6–6.9 mmol/L, 73 individuals) or IGT (2-h glucose 7.8–11.0 mmol/L, 27 individuals) or both IFG and IGT (36 individuals). To maximize power, those with diabetes and prediabetes were combined into a single dysglycemic group (total 164 participants) and compared to the 189 participants with normal glucose tolerance.

The study was approved by the Institutional Review Board at Cedars-Sinai Medical Center (protocol number Pro00046849, initially approved 28 December, 2016). All subjects gave written informed consent prior to participation.

### 4.2. Phenotyping Insulin Homeostasis

Insulin homeostasis traits herein refer to insulin secretion, insulin sensitivity, insulin secretion, and disposition index. To achieve the best balance of quality phenotyping without undue burden to participants, the oral glucose tolerance test (OGTT) was used to obtain measures of insulin homeostasis. Following an overnight fast, venous blood samples were obtained for the measurement of plasma glucose, insulin, and C-peptide before (fasting) and 30 and 120 min after the oral administration of a 75-g glucose load. While several OGTT-derived indices of insulin sensitivity/resistance have been developed, we utilized the Matsuda insulin sensitivity index (ISI). ISI is one of the indices most highly correlated with directly quantified (by euglycemic clamp) insulin sensitivity (r 0.7 to 0.8). [41] Furthermore, the Matsuda ISI can be calculated using fewer OGTT time points, without reduction in correlation with directly measured insulin sensitivity [48]. Our measure of insulin secretion is the area under curve (AUC) for insulin from baseline to 30 min over the corresponding AUC for glucose (AUC-Ins_30_/AUC-Glu_30_). This measure was found to be highly correlated with first phase insulin secretion from the intravenous glucose tolerance test (r = 0.7) [41]. In addition, this AUC-based insulin secretion measure has been found to have a hyperbolic relationship with insulin sensitivity, consistent with the relationship found when insulin secretion and insulin sensitivity are measured with gold standard physiologic tests [49]. This relationship allows calculation of the disposition index, the product of insulin secretion and insulin sensitivity (herein, DI_30_ = ISI × AUC-Ins_30_/AUC-Glu_30_), which represents an index of insulin secretion that accounts for its degree of compensation for insulin resistance. Insulin clearance is measured as the AUC of C-peptide over the AUC of insulin (AUC-Cpep/AUC-Ins), a commonly used index of hepatic insulin extraction given that the liver clears insulin but not C-peptide [17].

### 4.3. Statistical Analysis

To normalize the distribution for all statistical analyses, a rank-based inverse normal transformation using the constant provided by Blom [50,51] was applied to BMI, fasting glucose, glucose at 120 min, fasting insulin, fasting C-peptide, insulin sensitivity, insulin secretion, disposition index, and insulin clearance. However, median values (with interquartile range) are presented in Tables to facilitate interpretation. Phenotypes between groups were compared using Student’s *t*-tests (when comparing two groups, i.e., normal glucose tolerance vs. dysglycemia) and one-way ANOVA with Tukey’s HSD post hoc test (when comparing more than two groups, i.e., normal glucose vs. prediabetes vs. diabetes). Chi-squared tests were used to compare sex and race between groups. Linear regression was used to assess the correlations of insulin clearance with insulin sensitivity, insulin secretion, and disposition index.

A series of logistic regression models was constructed with dysglycemia as the dependent variable. To assess for multicollinearity (high degree of intercorrelation between independent variables) within the regression models, variance inflation factors (VIF) were calculated. VIF less than 2.5 are acceptable while VIF above 5 indicate a high degree of multicollinearity that can lead to inaccurately estimated regression coefficients for individual independent variables. In the base model, the independent variables were age, sex, BMI, and race. Additional models were constructed in which insulin sensitivity (model 1), insulin secretion (model 2), insulin clearance (model 3), and disposition index (model 4) were included as independent variables in separate models along with those of the base model. The independent variables of the penultimate model consisted of those of the base model plus insulin sensitivity, insulin secretion, and insulin clearance jointly (model 5). Given that some degree of inflation was observed in the latter model, a final model consisting of the base model plus disposition index and insulin clearance was calculated (model 6). Akaike’s information criterion (AIC) was included as a measure of model fit. A difference in AIC (∆AIC) < 2 between two models indicates equal support for both models, ∆AIC of 4–7 suggests moderate support for the model with the lower AIC, while ∆AIC > 10 indicates substantial support for the model with the lower AIC [52].

Area under the receiver operating characteristic curve (AUROC) was derived from the logistic regression models, to represent the trade-off between sensitivity and specificity for each model. Binormal curve smoothing was fitted to the quantiles of the sensitivities and specificities [53], with 95% confidence intervals estimated via bootstrapping (2000 permutations). Comparisons between the raw AUROCs for different models were calculated using the DeLong method for paired curves, with a two-tailed hypothesis [54].

In secondary analyses, we examined insulin homeostasis traits, carried out the regression modeling outlined above, and compared the corresponding AUROC values separately in African Americans and non-Hispanic Whites with and without dysglycemia. We also conducted these analyses in subgroups with and without prediabetes (IGT and/or IFG), IGT, or IFG.

## Figures and Tables

**Figure 1 metabolites-11-00420-f001:**
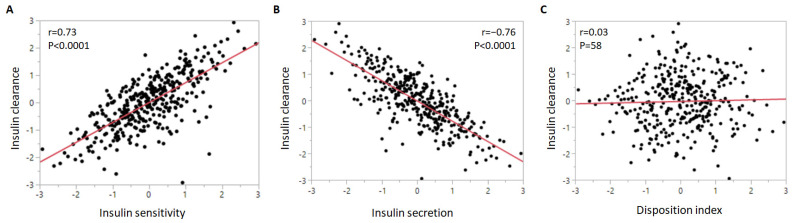
Correlation of insulin clearance with (**A**) insulin sensitivity, (**B**) insulin secretion, and (**C**) disposition index. The insulin homeostasis traits were inverse normal transformed.

**Figure 2 metabolites-11-00420-f002:**
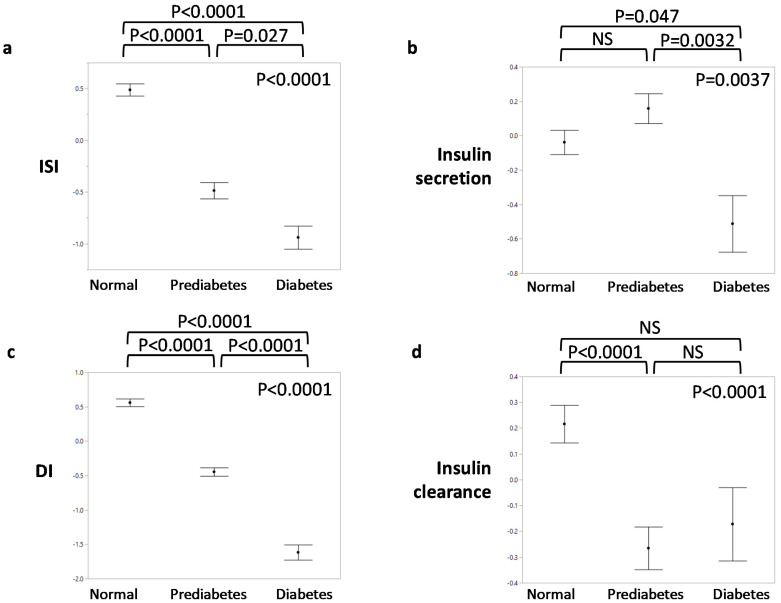
Insulin homeostasis traits by glycemic status. (**a**) Insulin sensitivity index (ISI); (**b**) insulin secretion; (**c**) disposition index (DI); (**d**) insulin clearance. The insulin homeostasis traits were inverse normal transformed. *p*-values in the upper right corner of each graph indicate overall significance by ANOVA. *p*-values above each graph are for pairwise comparisons by Tukey’s HSD post hoc test.

**Figure 3 metabolites-11-00420-f003:**
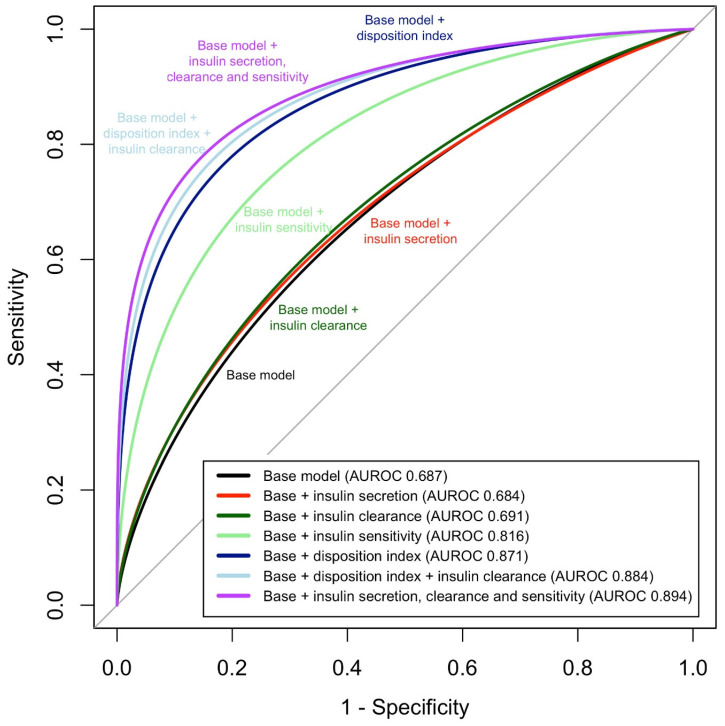
Area under the receiver operating characteristics curve (AUROC) for models examining the sensitivity and specificity of four insulin homeostasis traits to discriminate dysglycemia from normal glucose tolerance, in separate models and for models combining traits.

**Table 1 metabolites-11-00420-t001:** Clinical and glucose and insulin homeostasis characteristics by glycemic category.

	Normal Glucose Tolerance (*n* = 189)	Prediabetes (*n* = 136)	Diabetes (*n* = 28)	*p*-Value †
Age	58.0 (13.0) *	61.0 (15.0)	65.0 (12.3)	0.0024
Sex (% male)	32.2% *	45.6%	50%	0.013
Race (% African American)	34.4%	40.4%	32.1%	0.47
BMI (kg/m^2^)	26.0 (7.2) *	28.8 (7.7)	31.7 (7.0)	<0.0001
Waist-to-hip ratio	0.90 (0.10) *	0.95 (0.12) **	1.01 (0.085) ***	<0.0001
Systolic blood pressure (mmHg)	117.5 (25.8)	120.8 (24.4)	122.5 (15.3)	0.20
Diastolic blood pressure (mmHg)	71.5 (14.8)	72.0 (12.4)	73.5 (16.3)	0.55
Fasting glucose (mmol/L)	5.16 (0.39) *	5.77 (0.50) **	6.22 (1.25) ***	<0.0001
Fasting insulin (pmol/L)	43.2 (33.6) *	70.2 (56.4)	98.4 (63.6)	<0.0001
Glucose at 120 min (mmol/L)	5.61 (1.83) *	7.44 (2.71) **	12.07 (1.97) ***	<0.0001
Fasting C-peptide (nmol/L)	0.58 (0.29) *	0.84 (0.46) **	1.05 (0.62) ***	<0.0001
Insulin sensitivity (ISI)	5.57 (4.59) *	3.08 (2.32) **	2.05 (1.12) ***	<0.0001
Insulin secretion (AUC-Ins_30_/AUC-Glu_30_)	0.35 (0.29)	0.38 (0.39)	0.23 (0.27) *	0.0037
Disposition index (DI_30_)	1.93 (1.10) *	1.10 (0.67) **	0.59 (0.24) ***	<0.0001
Insulin clearance (AUC-Cpep/AUC-Ins) ‡	0.11 (0.054)	0.093 (0.045)	0.10 (0.024)	<0.0001

Data are median (interquartile range) for quantitative traits and *n* (%) for sex and race. ISI, insulin sensitivity index; AUC-Ins_30_/AUC-Glu_30_, insulin secretion; DI_30_, disposition index; AUC-Cpep/AUC-Ins, insulin clearance. Asterisks (*, **, or ***) indicate values that are significantly different from the other(s) in each row by Tukey’s HSD post hoc test or chi square test (sex, race). † *p*-value for each row by ANOVA or chi square test. ‡ For the insulin clearance variable, the value was significantly different only between the normal glucose tolerance group and the prediabetes group.

**Table 2 metabolites-11-00420-t002:** Insulin homeostasis traits in individuals with normal glucose tolerance and dysglycemia.

	Normal Glucose Tolerance (*n* = 189)	Dysglycemia (*n* = 164)	*p*-Value
Insulin sensitivity	5.57 (4.59)	2.67 (2.23)	<0.0001
Insulin secretion	0.35 (0.29)	0.36 (0.34)	0.44
Disposition index	1.93 (1.10)	1.01 (0.68)	<0.0001
Insulin clearance	0.11 (0.054)	0.095 (0.043)	<0.0001

Data are median (interquartile range).

**Table 3 metabolites-11-00420-t003:** Parameter estimates and area under the receiver operating characteristics curve for logistic regression models examining the association of four insulin homeostasis traits with dysglycemia, in separate models and for models combining traits.

	Base Model	Model 1	Model 2	Model 3	Model 4	Model 5	Model 6
Age	0.045 *** (0.013)	0.019 (0.015)	0.044 *** (0.013)	0.042 *** (0.013)	0.002 (0.017)	−0.012 (0.018)	−0.007 (0.017)
Sex (male)	0.611 *** (0.231)	0.218 (0.271)	0.649 *** (0.234)	0.542 *** (0.235)	0.635 ** (0.289)	0.341 (0.306)	0.546 * (0.298)
BMI	0.500 *** (0.125)	−0.276 * (0.162)	0.557 *** (0.135)	0.377 *** (0.134)	0.127 (0.157)	−0.410 ** (0.197)	−0.146 (0.177)
Race (African American)	0.039 (0.248)	−0.181 (0.268)	0.106 (0.254)	−0.187 (0.265)	0.808 ** (0.327)	0.504 (0.361)	0.377 (0.349)
Insulin sensitivity		−1.657 *** (0.212)				−3.413 *** (0.403)	
Insulin secretion			−0.158 (0.128)			−1.925 *** (0.304)	
Insulin clearance				−0.359 *** (0.139)		0.129 (0.299)	−0.696 *** (0.182)
Disposition index					−2.066 *** (0.235)		−2.226 *** (0.254)
AUROC	0.678 (0.625–0.734)	0.816 (0.772–0.863)	0.684 (0.631–0.739)	0.691 (0.635–0.745)	0.871 (0.833–0.90)	0.894 (0.860–0.925)	0.884 (0.849–0.918)
AIC	461.100	375.940	461.583	456.234	324.260	302.540	310.432

* *p* < 0.05; ** *p* < 0.01; *** *p* < 0.001. Parameter estimates are listed with standard error in parentheses. AIC, Akaike’s information criterion; AUROC, area under the receiver operating characteristic curve. AUROC values were smoothed using binormal smoothing (displayed in Figure 1). BMI and the insulin homeostasis traits were inverse normal transformed.

**Table 4 metabolites-11-00420-t004:** Comparisons of AUROC values from different logistic regression models for dysglycemia.

	Base Model	Model 1	Model 2	Model 3	Model 4	Model 5	Model 6
Base model	-	−5.23	−0.78	−0.80	−6.58	−7.39	−7.15
Model 1: base + insulin sensitivity	1.67 × 10^−7^	-	4.59	5.97	−1.85	−4.23	−2.94
Model 2: base + insulin secretion	0.44	4.52 × 10^−6^	-	−0.30	−6.82	−7.20	−7.13
Model 3: base + insulin clearance	0.42	2.41 × 10^−9^	0.76	-	−5.94	−7.35	−7.05
Model 4: base + disposition index	4.62 × 10^−11^	0.065	9.20 × 10^−12^	2.78 × 10^−9^	-	−2.00	−1.71
Model 5: base + insulin sensitivity, secretion, clearance	1.44 × 10^−13^	2.35 × 10^−5^	6.26 × 10^−13^	2.02 × 10^−13^	0.046	-	1.35
Model 6: base + insulin clearance, disposition index	9.02 × 10^−13^	0.003	1.01 × 10^−12^	1.79 × 10^−12^	0.087	0.18	-

Values above the diagonal (upper right) are the Z scores. Values below the diagonal (lower left) are the *p*-values.

## Data Availability

The data are not publicly available because participants did not give consent for the data to be publicly posted. Interested researchers should contact the corresponding author and submit their credentials to the Cedars-Sinai Institutional Review Board for determination of whether if they are eligible to have access to study data. Upon approval, a limited dataset necessary for replication would be provided to the investigator.

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
