# Peer review of "Defining the Relative Role of Insulin Clearance in Early Dysglycemia in Relation to Insulin Sensitivity and Insulin Secretion: The Microbiome and Insulin Longitudinal Evaluation Study (MILES)"

_metabolites, 2021, doi:10.3390/metabo11070420_

Round 1

Reviewer 1 Report

REVIEW

Defining the relative role of insulin clearance in early dysglycemia in relation to insulin sensitivity and insulin secretion: the microbiome and insulin longitudinal evaluation study (Miles).

Wood et al.

The authors studied several measures of insulin homeostasis mainly using OGTT as a challenge in a cohort of 353 individuals (129 African Americans, 224 non-Hispanic Whites).   The main conclusion is that insulin clearance was not an independent predictor of dysglycemia (pooled prediabetes and diabetes individuals) whereas both insulin secretion and sensitivity were predictors.   I have a few queries for the authors.

I agree with the author’s premise that it is important to consider multiple aspects of insulin homeostasis.  This concept can be taken even further if one also includes analyses of glucagon or other peptides that influence blood glucose levels.   Do you have data on other peptides that contribute to blood glucose ?

I get why the authors pooled prediabetic and diabetic individuals.  However, presumably the prediabetic folks would likely be at various stages in their path to diabetes and hence this is a rather heterogeneous group.  Would this variability have a confounding effect on your analyses?   You have acknowledged this by including Table 1 and Fig. 2 which is perfectly acceptable.   I realize this is a difficult proposition.  However, is there a way that you might have a measure of heterogeneity and also acknowledge that the really important comparison is with prediabetic patients versus normal?

Also, you are pooling whites and African Americans.   This seems a bit counter intuitive as you state on p. 6, line 210-211 the following:   “ When  208 stratified by race, the same differences in insulin homeostasis traits were observed, ex- 209 except that insulin clearance was not significantly different between African Americans  with and without dysglycemia”.   Would it be beneficial to include the data split on the basis of race in an appendix?

Beta cell failure is likely a major contributor to T2D.  Could ISI be used as a measure of beta cell mass similar to the crude measure of height of the first phase insulin peak during IVGTT ?  Along similar lines, is it possible to use C-peptide/insulin as a measure of differences in unfolded protein response which may be inhibited as patients progress to T2D?

Why is the % males so low in your control group?   Is this a problem.

In Table 1 you give measures in International units for glucose but in your description of patient groups, you use mass concentration, pick one.

Do you have data on glucagon, PP, GLP-1, etc. concs or any measures of liver function?

At the bottom of Table 4 there is a statement:  “Values above the diagonal are the Z scores. Values below the diagonal are the P values. “   Is this meant to refer to Fig. 3 ?  There is no diagonal in Table 4.

I was not aware of the term VIF.  Maybe consider adding a one sentence definition.

Overall, a nice study that makes an important contribution.  Thank you for this work.

Author Response

The authors studied several measures of insulin homeostasis mainly using OGTT as a challenge in a cohort of 353 individuals (129 African Americans, 224 non-Hispanic Whites).   The main conclusion is that insulin clearance was not an independent predictor of dysglycemia (pooled prediabetes and diabetes individuals) whereas both insulin secretion and sensitivity were predictors.   I have a few queries for the authors.

  1. I agree with the author’s premise that it is important to consider multiple aspects of insulin homeostasis.  This concept can be taken even further if one also includes analyses of glucagon or other peptides that influence blood glucose levels.   Do you have data on other peptides that contribute to blood glucose ?

Response: At this time, we do not have data on additional peptides such as glucagon or incretin hormones. We hope to generate such data in the future.

  1. I get why the authors pooled prediabetic and diabetic individuals.  However, presumably the prediabetic folks would likely be at various stages in their path to diabetes and hence this is a rather heterogeneous group.  Would this variability have a confounding effect on your analyses?   You have acknowledged this by including Table 1 and Fig. 2 which is perfectly acceptable.   I realize this is a difficult proposition.  However, is there a way that you might have a measure of heterogeneity and also acknowledge that the really important comparison is with prediabetic patients versus normal?

Response: We pooled the prediabetic and diabetic individuals to maximize the power of our analyses. Because those with diabetes were not known to have diabetes until they were enrolled in the study, their diabetes is considered to be mild, newly diagnosed diabetes. Therefore, we have used the term “early dysglycemia” as prediabetes and newly diagnosed diabetes should lie close together on the spectrum of diabetes. We believe that readers will be interested in early dysglycemia. Another reason to focus on dysglycemia rather than prediabetes is that similar papers examining insulin clearance either examined dysglycemia (the PROMISE study, ref 17) or incident diabetes (IRAS Family Study [ref 9], study in Southwest Native Americans [ref 13]), outcomes comparable to early dysglycemia. In the paper (lines 127, 132, and 366-367), we acknowledged that our analyses mainly reflect prediabetes, as the dysglycemic group consists mainly of people with prediabetes (83% of the group). To address the issue of heterogeneity raised by this and other reviewers, we have included in supplementary data a series of sub-analyses of more homogeneous groups, including those with prediabetes (excluding those with diabetes) (Supplemental Tables 3, 6, 7), those with those with impaired glucose tolerance (Supplemental Tables 4, 8, 9), and those with impaired fasting glucose (Supplemental Tables 5, 10, 11). These secondary analyses will satisfy readers who wish to see results in these subgroups. Results were generally similar, with very few differences noted (lines 132-138, 190-191).

  1. Also, you are pooling whites and African Americans.   This seems a bit counter intuitive as you state on p. 6, line 210-211 the following: “ When  208 stratified by race, the same differences in insulin homeostasis traits were observed, ex- 209 except that insulin clearance was not significantly different between African Americans  with and without dysglycemia”.   Would it be beneficial to include the data split on the basis of race in an appendix?

Response: We now include in supplemental material (Supplementary Tables 1-2 and 12-15) analyses stratified by race (described in lines 131, 192-200, 459-461). While we did make note of the difference cited above, in numerous other analyses, whites and African Americans displayed the same results. Most importantly, in the regression models and AUROC analysis on which the main conclusions of the paper are based, results were quite similar between the groups.

  1. Beta cell failure is likely a major contributor to T2D.  Could ISI be used as a measure of beta cell mass similar to the crude measure of height of the first phase insulin peak during IVGTT?  Along similar lines, is it possible to use C-peptide/insulin as a measure of differences in unfolded protein response which may be inhibited as patients progress to T2D?

Response: ISI is regarded as a measure of insulin sensitivity, validated with high correlation to insulin sensitivity measured directly by euglycemic clamp or frequently sampled IV glucose tolerance test. Regarding beta cell function, we have included both a measure of acute insulin response as well as the disposition index. The disposition index is considered one of the best markers of beta cell function, as it reflects the appropriateness of beta cell response to insulin resistance. In line with prior studies, we found that disposition index was highly predictive of dysglycemia. We agree that C-peptide/insulin ratio, while used by many as a marker of insulin clearance, might also reflect insulin processing. Given that assessment of unfolded protein response would require molecular studies beyond the scope of the current clinical research, we have addressed this briefly with a new statement in the Discussion (lines 359-360).

  1. Why is the % males so low in your control group?   Is this a problem.

Response: For some reason, men were less interested in participating in the study than women. We did our best to ameliorate this by targeting recruitment efforts towards men (e.g., presenting the study at men’s clubs). The cohort is composed of 38% men. The lower proportion of these men in the control group likely reflects a higher prevalence of dysglycemia in men than women. This is why we included sex as a covariate in all of our regression models.

  1. In Table 1 you give measures in International units for glucose but in your description of patient groups, you use mass concentration, pick one.

Response: We now uniformly use SI units as required by the journal. Changes were made in lines 386-393.

  1. Do you have data on glucagon, PP, GLP-1, etc. concs or any measures of liver function?

Response: Such measures are not currently available.

  1. At the bottom of Table 4 there is a statement:  “Values above the diagonal are the Z scores. Values below the diagonal are the P values. “   Is this meant to refer to Fig. 3 ?  There is no diagonal in Table 4.

Response: The “diagonal” in Table 4 are the cells that contain no data, which form a diagonal in the table. We have clarified this by modifying the Table legend (line 167) as follows: “Values above the diagonal (upper right) are the Z scores. Values below the diagonal (lower left) are the P values.”

  1. I was not aware of the term VIF.  Maybe consider adding a one sentence definition.

Response: VIF is defined with the following text in the Statistical Analysis section (lines 439-441): “To assess for multicollinearity within the regression models, variance inflation factors (VIF) were calculated. VIF less than 2.5 are acceptable while VIF above 5 indicate poorly estimated regression coefficients due to multicollinearity.”

  1. Overall, a nice study that makes an important contribution.  Thank you for this work.

Response: Thank you for recognizing the importance of this working and making suggestions that increased the quality of the paper.

Reviewer 2 Report

The authors nicely show the utility of using an OGTT that measures glucose, insulin, and c-peptide in order to quantify insulin resistance, insulin secretion, insulin clearance and disposition index. From these metrics, they developed various models that used these metrics to see which were the most predictive of dysglycemia in the race groups studied. They found that insulin resistance and insulin secretion are the greatest predictors of dysglycemia. Insulin clearance can also be an indicator of dysglycemia but is not found to be a significant predictor of dysglycemia when using all three metrics specifically in this race group. I think the study nicely supports findings from other studies and adds to the knowledge of field to support patients with prediabetes.

The follow up study in 5 years when they will measure the same metrics again will be very interesting to see how their findings in dysglycemia translate to the progression of type II diabetes. It would be nice if then they can find correlations but also identify ranges for these tests to identify patients at more risk, and when intervention can make the most difference.

Author Response

The authors nicely show the utility of using an OGTT that measures glucose, insulin, and c-peptide in order to quantify insulin resistance, insulin secretion, insulin clearance and disposition index. From these metrics, they developed various models that used these metrics to see which were the most predictive of dysglycemia in the race groups studied. They found that insulin resistance and insulin secretion are the greatest predictors of dysglycemia. Insulin clearance can also be an indicator of dysglycemia but is not found to be a significant predictor of dysglycemia when using all three metrics specifically in this race group. I think the study nicely supports findings from other studies and adds to the knowledge of field to support patients with prediabetes.

The follow up study in 5 years when they will measure the same metrics again will be very interesting to see how their findings in dysglycemia translate to the progression of type II diabetes. It would be nice if then they can find correlations but also identify ranges for these tests to identify patients at more risk, and when intervention can make the most difference.

Response: We thank the reviewer for their supportive comments. We concur that future prospective analyses in this cohort will be of great interest.

Reviewer 3 Report

In this study, authors studied OGTT data for insulin sensitivity, insulin secretion, DI  and insulin clearance in a small group of normal glycaemic and dysglycemic people. The authors concluded that insulin clearance does not predict the development of dysglycaemia but insulin sensitivity and secretion does. 

The study is well written with clear statistical analysis. The results of the study are expected as reported by multiple studies that insulin secretion and sensitivity are the major predictors of T2D. 

Do the authors want to assess the differnece between normal vs prediabetes or diabetes? if it is prediabetes, T2D cases should not be included.  The authors should pick one group and stick to it.  This is because the authors showed in the current study that diabetes cases have severe defects. Thus including the T2D cases with prediabetes cases, the authors have inflated the difference between the normal and dysglycemic people. Thus making the logistic regression analysis biased.

The study lacks the sensitivity analysis for prediabetes cases only. This also should be analysed for people with only IFG, only  IGT or both together as there are data showing the differences between these three groups. 

However, If they want to show that ISI, secretion clearance have a continuum effect from normal to prediabetes to diabetes (better approach to answer the study question particularly as they see the stepwise decline in the parameters), They should use linear regression to show that lack of effect of insulin clearance on the transition from normal to prediabetes to diabetes.

C-peptide and insulin also excreted by the kidney. Did authors assess how kidney function ( i.e GFR)  affected their results? 

The authors suggested using derived data to predict dysglycaemia (which one). Did they use HbA1c in a model to show that these OGTT derived measures are better or still added on HbA1c? OGTT is has a huge  CV  along with difficulties in conducting them in primary care making it less useful in day to day use. 

Author Response

In this study, authors studied OGTT data for insulin sensitivity, insulin secretion, DI  and insulin clearance in a small group of normal glycaemic and dysglycemic people. The authors concluded that insulin clearance does not predict the development of dysglycaemia but insulin sensitivity and secretion does. 

The study is well written with clear statistical analysis. The results of the study are expected as reported by multiple studies that insulin secretion and sensitivity are the major predictors of T2D. 

  1. Do the authors want to assess the difference between normal vs prediabetes or diabetes? if it is prediabetes, T2D cases should not be included.  The authors should pick one group and stick to it.  This is because the authors showed in the current study that diabetes cases have severe defects. Thus including the T2D cases with prediabetes cases, the authors have inflated the difference between the normal and dysglycemic people. Thus making the logistic regression analysis biased.

Response: We pooled the prediabetic and diabetic individuals to maximize the power of our analyses. Because those with diabetes were not known to have diabetes until they were enrolled in the study, their diabetes is considered to be mild, newly diagnosed diabetes. Therefore, we have used the term “early dysglycemia” as prediabetes and newly diagnosed diabetes should lie close together on the spectrum of diabetes. We believe that readers will be interested in early dysglycemia. Another reason to focus on dysglycemia rather than prediabetes is that similar papers examining insulin clearance either examined dysglycemia (the PROMISE study, ref 17) or incident diabetes (IRAS Family Study [ref 9], study in Southwest Native Americans [ref 13]), outcomes comparable to early dysglycemia. In the paper (lines 127, 132, and 366-367), we acknowledged that our analyses mainly reflect prediabetes, as the dysglycemic group consists mainly of people with prediabetes (83% of the group). To address the issue of heterogeneity raised by this and other reviewers, we have included in supplementary data a series of sub-analyses of more homogeneous groups, including those with prediabetes (excluding those with diabetes) (Supplemental Tables 3, 6, 7), those with those with impaired glucose tolerance (Supplemental Tables 4, 8, 9), and those with impaired fasting glucose (Supplemental Tables 5, 10, 11). These secondary analyses will satisfy readers who wish to see results in these subgroups. Results were generally similar, with very few differences noted (lines 132-138, 190-191).

  1. The study lacks the sensitivity analysis for prediabetes cases only. This also should be analysed for people with only IFG, only IGT or both together as there are data showing the differences between these three groups. 

Response: As noted above, we have added analyses based on subsets of participants, including those with IFG and those with IGT. We did not include analyses of those with both IFG plus IGT because that group numbers only 36 individuals, which would result in unstable estimates. In general, the results in these subgroups were quite similar to the results of main analysis on dysglycemia.

  1. However, If they want to show that ISI, secretion, clearance have a continuum effect from normal to prediabetes to diabetes (better approach to answer the study question particularly as they see the stepwise decline in the parameters), They should use linear regression to show that lack of effect of insulin clearance on the transition from normal to prediabetes to diabetes.

Response: In Table 1 and Figure 1 we presented data according to three states of glucose homeostasis (normal, prediabetes, diabetes) as descriptive results that would be of interest to readers. However, to maximize power, our main analyses are based on the binary analysis (normal vs dysglycemia). The relatively low number of individuals with diabetes (n=28) would reduce the reliability of analyses where these individuals form a separate group. We hope the reviewer accept our justification for combining those with newly discovered, untreated diabetes with those with prediabetes as ‘early dysglycemia.’ We have added extensive sensitivity analyses, as requested by this reviewer and others, that will give readers a detailed view of how dysglycemia performs compared to prediabetes, IFG, or IGT.

  1. C-peptide and insulin also excreted by the kidney. Did authors assess how kidney function (i.e GFR) affected their results?

Response: We currently do not have measures of kidney function in our data.

  1. The authors suggested using derived data to predict dysglycaemia (which one). Did they use HbA1c in a model to show that these OGTT derived measures are better or still added on HbA1c? OGTT is has a huge CV along with difficulties in conducting them in primary care making it less useful in day to day use.

Response: We appreciate this suggestion. However, we cannot address it because we do not have HbA1c available in our data.

Reviewer 4 Report

Wood et al carried out a cross-sectional analysis, based on the baseline data of MILES prospective studies on a cohort of 353 non-hispanic whites and African-Americans (AA) to investigate the role of insulin clearance as an independent predictor of early hyperglycemia. Insulin clearance was assessed as AUC C-peptide/AUC Insulin during an oral glucose tolerance test, instead of the gold standard of assessing it during a hyperinsulinemic-euglycemic clamp analysis or FSIGT. The authors concluded that insulin clearance may play a significant role in the early dysglycemic state, but that insulin secretion and insulin resistance combined play a more causal role as diabetes progresses.  In this regard, do the authors wish to rephrase the last sentence in the abstract?

Minor concerns over the limited cohort (353 subjects) of mixed race [non-Hispanic whites and African Americans (AA) while earlier studies showing reduced insulin extraction in AA]. The authors stated that the age range included in the studies is 40-80 years of age, but the table states ages of 58-65. Because insulin clearance may progressively decline in age in parallel to insulin secretion, it is recommended that the real range of age of the subjects used in the current studies will be stated. 

Nonetheless, the studies are well controlled and the data interpretation is solid. The discussion is extraordinary comprehensive and would constitute a great introduction to this area of research. Enthusiasm is high.

Author Response

Wood et al carried out a cross-sectional analysis, based on the baseline data of MILES prospective studies on a cohort of 353 non-hispanic whites and African-Americans (AA) to investigate the role of insulin clearance as an independent predictor of early hyperglycemia. Insulin clearance was assessed as AUC C-peptide/AUC Insulin during an oral glucose tolerance test, instead of the gold standard of assessing it during a hyperinsulinemic-euglycemic clamp analysis or FSIGT.

  1. The authors concluded that insulin clearance may play a significant role in the early dysglycemic state, but that insulin secretion and insulin resistance combined play a more causal role as diabetes progresses. In this regard, do the authors wish to rephrase the last sentence in the abstract?

Response: What we consider to be a limitation of many past studies of insulin clearance is that it typically was analyzed in isolation. Given the inter-relationships between insulin resistance, insulin secretion, and insulin clearance, we feel that joint analysis is critical to dissecting which defects are independently associated with risk of dysglycemic progression. In our joint analysis (Model 5) highlighted in the abstract, only insulin resistance and reduced insulin secretion were associated with dysglycemia. As insulin clearance was not significantly associated with early dysglycemia in that analysis, we feel that the abstract as currently written best reflects our results.

  1. Minor concerns over the limited cohort (353 subjects) of mixed race [non-Hispanic whites and African Americans (AA) while earlier studies showing reduced insulin extraction in AA]. The authors stated that the age range included in the studies is 40-80 years of age, but the table states ages of 58-65. Because insulin clearance may progressively decline in age in parallel to insulin secretion, it is recommended that the real range of age of the subjects used in the current studies will be stated. 

Response: We now include supplemental data (Supplemental Tables 1-2, 12-15) showing results stratified by race. Results between the two race groups were generally similar. The age range of participants in the study is indeed 40-80. Table 1 lists the median ages of 58, 61, and 65 for those with normal glucose tolerance, prediabetes, and diabetes, respectively. Interquartile ranges are included in Table 1 to give readers a sense of the spread of age in each category.

Nonetheless, the studies are well controlled, and the data interpretation is solid. The discussion is extraordinary comprehensive and would constitute a great introduction to this area of research. Enthusiasm is high.

Round 2

Reviewer 3 Report

The authors have appropriately addressed my concerns. Thank you.

Author Response

Thank you